# Acceptance of and Adherence to a Four-Dose RTS,S/AS01 Schedule: Findings from a Longitudinal Qualitative Evaluation Study for the Malaria Vaccine Implementation Programme

**DOI:** 10.3390/vaccines11121801

**Published:** 2023-12-01

**Authors:** Jessica Price, Nikki Gurley, Margaret Gyapong, Evelyn Korkor Ansah, Kofi Awusabo-Asare, Samuel Fosu Gyasi, Pearson Nkhoma, Alinane Linda Nyondo-Mipando, George Okello, Jayne Webster, Nicola Desmond, Jenny Hill, W. Scott Gordon

**Affiliations:** 1PATH, 2201 Westlake Ave, Seattle, WA 98102, USA; sgordon@path.org; 2King County Department of Community and Human Services, 401 5th Ave #500, Seattle, WA 98104, USA; nikki.gurley@gmail.com; 3Institute of Health Research, University of Health and Allied Sciences, PMB 31, Ho, Volta Region, Ghana; mgyapong@uhas.edu.gh; 4Center for Malaria Research, University of Health and Allied Sciences, PMB 31, Ho, Volta Region, Ghana; eansah@uhas.edu.gh; 5Department of Population and Health, University of Cape Coast, New Administration Block, Cape Coast, Ghana; k.awusabo-asare@ucc.edu.gh; 6Center for Research in Applied Biology, School of Sciences, University of Energy and Natural Resources, P.O. Box 214, Sunyani, Ghana; samuel.gyasi.fosu@gmail.com; 7Goldsmiths, University of London, 8 Lewisham Way, New Cross, London SE14 6NW, UK; p.nkhoma@gold.ac.uk; 8Department of Health Systems and Policy, Kamuzu University of Health Sciences, Private Bag 360, Blantyre 3, Malawi; lmipando@kuhes.ac.mw; 9Kenya Medical Research Institute, Centre for Geographic Medicine, Kisumu P.O. Box 1578, Kenya; gokello2002@gmail.com; 10London School of Hygiene and Tropical Medicine, Keppel St, London WC1E 7HT, UK; jayne.webster@lshtm.ac.uk; 11Department of International Public Health, Liverpool School of Tropical Medicine, University of Liverpool, Pembroke Pl, Liverpool L3 5QA, UK; nicola.desmond@lstmed.ac.uk; 12Department of Clinical Sciences, Liverpool School of Tropical Medicine, Pembroke Pl, Liverpool L3 5QA, UK; jenny.hill@lstmed.ac.uk

**Keywords:** malaria vaccine pilot evaluation, vaccine acceptance, vaccine trust, vaccine uptake barriers

## Abstract

Background: The WHO recommended the use of the RTS,S/AS01 malaria vaccine (RTS,S) based on a pilot evaluation in routine use in Ghana, Kenya, and Malawi. A longitudinal qualitative study was conducted to examine facilitators and barriers to uptake of a 4-dose RTS,S schedule. Methods: A cohort of 198 caregivers of RTS,S-eligible children from communities where RTS,S was provided through the pilot were interviewed three times over a ≈22-month, 4-dose schedule. The interviews examined caregiver perceptions and behaviors. Children’s vaccination history was obtained to determine dose uptake. Results: 162 caregivers remained at round 3 (R3); vaccination history was available for 152/162 children. Despite early rumors/fears, the uptake of initial doses was high, driven by vaccine trust. Fears dissipated by R2, replaced with an enthusiasm for RTS,S as caregivers perceived its safety and less frequent and severe malaria. By R3, 98/152 children had received four doses; 34 three doses; 9 one or two doses; and 11 zero doses. The health system and information barriers were important across all under-dose cases. Fears about AEFIs/safety were important in zero-, one-, and two-dose cases. Competing life/livelihood demands and complacency were found in three-dose cases. Regardless of the doses received, caregivers had positive attitudes towards RTS,S by R3. Conclusions: Findings from our study will help countries newly introducing the vaccine to anticipate and preempt reasons for delayed acceptance and missed RTS,S doses.

## 1. Introduction

The World Health Organization’s (WHO) Malaria Report 2022 indicates steady increases in cases globally since 2016 [1]. An estimated 593,000 malaria deaths occurred in 2021, 95% in the WHO African Region and 79% in children under five. Malaria vaccines are important additions to existing prevention and control interventions [2]. A phase 3 clinical trial of the RTS,S/AS01 vaccine (RTS,S) demonstrated its efficacy against clinical and severe malaria in children following a four-dose schedule starting at five months [3]. In October 2021 the WHO recommended the use of RTS,S in children in areas with moderate-to-high malaria transmission as part of comprehensive malaria interventions [4].

The WHO recommendation was also based on evidence from the pilot introduction of RTS,S to assess its safety, effectiveness, and feasibility when implemented through routine immunization programs. Pilots of the Malaria Vaccine Implementation Programme were launched in 2019 in parts of Ghana, Kenya, and Malawi [5], involving subnational introduction in areas with moderate-to-high malaria transmission concurrently with studies evaluating the vaccine in routine use. Led by the ministries of health in each country, the program was coordinated by WHO in collaboration with PATH, UNICEF, and other partners, with doses donated by GSK, the manufacturer. Of note, the pilots were underway when COVID-19 lockdowns and other safety measures disrupted routine immunization services in sub-Saharan Africa [6,7].

A qualitative study—the Healthcare Utilization Study (HUS)—was conducted in all three countries as a component of the pilot evaluations. To complement behavioral findings from representative household surveys [5] and generate both globally relevant and country-specific insights, the HUS was designed to understand (i) factors that promote or obstruct RTS,S adoption and adherence to the four-dose schedule; (ii) the potential impact of RTS,S uptake on other malaria prevention and treatment behaviors; and (iii) RTS,S delivery challenges. This paper presents the findings from a cohort of child caregivers, focusing on objective (i): factors promoting or obstructing uptake.

The HUS was conducted by a consortium of partners led by the University of Health and Allied Sciences in Ghana, the Liverpool School of Tropical Medicine in Kenya, and the Malawi-Liverpool Wellcome Trust in Malawi; PATH led the overall study (Appendix A). Country partners led country-specific studies, and collected and processed data; represented the project vis à vis local authorities; and contributed to cross-country analyses. PATH analyzed cross-country data, represented the project vis à vis global partners, and coordinated the consortium. Ethical approvals were obtained from Ghana’s Health Service Ethics Review Committee, the University of Malawi College of Medicine’s Research and Ethics Committee, the Kenya Medical Research Institute’s Scientific and Ethics Unit, the Liverpool School of Tropical Medicine Research Ethics Committee, the London School of Hygiene & Tropical Medicine’s Observational/Interventions Research Ethics Committee, and PATH’s Research Ethics Committee. Written informed consent was obtained from every study participant.

## 2. Study Design and Methods

The HUS was designed as a qualitative descriptive study, unconstrained by theoretical and philosophical commitments [8,9,10]. In taking this naturalistic stance, the intent was to provide a “low-inference… straight description” [8] of the issues promoting or obstructing the uptake of a four-dose RTS,S schedule. Common site selection and participant enrollment procedures were used to ensure comparability across individuals, sites, and countries, and to enhance the applicability of findings to contexts beyond the study sites [11,12,13]. A longitudinal approach [14,15] was adopted to understand the factors influencing RTS,S uptake over the 22+-month schedule, with data collected at critical points in the four-dose cycle. Reflecting the known and assumed factors likely to influence RTS,S adoption and adherence, a logic model guided the development of interview guides and analytic focus. (Appendix A).

### 2.1. Study Sites

The pilot was implemented in 158 pilot clusters—66 districts in Ghana, 46 sub-counties in western Kenya, and 46 immunization clinics/catchment areas in Malawi—that were randomized into RTS,S introduction or comparator areas [16]. Within the clusters in which RTS,S was introduced, 27 community sites (9/country) were selected for inclusion in the HUS. Defined as the smallest administrative unit[s] aligned with the services provided by at least one health facility, HUS communities were selected following purposeful criteria to achieve variation in socio-cultural, urban/rural, and immunization coverage contexts (Figure 1).

### 2.2. Child Caregiver Cohort

Within 1–2 months after RTS,S introduction, caregivers were selected from households in which at least one child living there was eligible to receive RTS,S dose 1 (RTS,S-1) (Figure 2) [16]. Households in Ghana and Malawi were identified using WHO’s EPI Sampling Technique (EPI Coverage Survey, Expanded Programme on Immunization, WHO (pp. 15–18)), modified to ensure selected households were sufficiently spaced. In Kenya, household selection was based on lists of eligible children compiled by community health workers. When multiple RTS,S-eligible children were found to be living in the same household, a random selection procedure was used to select the caregiver to invite to the study. Individuals who were ≥15 years old and identified by household members as the person primarily responsible for care of the child were included. Target enrollment at round 1 (R1) was seven caregivers per study site, 63 per country, and 189 total. Investigators in Kenya replaced caregivers lost-to-follow-up (LTFU) in R2; LTFU cases were not replaced in Ghana and Malawi.

### 2.3. Data Collection

Participants were interviewed individually three times (Figure 2). At each round, data were collected on the caregiver’s socio-demographics (Appendix A) and the child’s vaccination history, extracted from their child health card (Appendix A). Semi-structured interviews (≈45–60 min) were also conducted at each round (Appendix A). R1 interviews focused on caregivers’ exposure to RTS,S messages; initial perceptions, concerns, and questions about the vaccine; experiences at the most recent vaccination visit, including information received about RTS,S; and behaviors related to malaria prevention and treatment. Focusing on changes in attitudes and behaviors, R2 interviews were conducted mid-way through the ≈22-month schedule. Though slightly delayed due to COVID closures, R2 interviews were conducted when children should have received three RTS,S doses (Figure 2). R3 was conducted when the child was around 24 months old and should have completed the schedule. R3 questions focused on the reasons the child received all, some, or no RTS,S doses and on caregivers’ perceptions about the vaccine’s impact on malaria frequency and severity. Interview guides were developed to reflect key knowledge needs for the pilot evaluations and based on the collective experience and expertise of study investigators.

Interviews were conducted in private locations and in participants’ languages or English, if preferred. Socio-demographic and vaccination history data were recorded on structured sheets or entered directly into handheld devices. Semi-structured interviews were audio-recorded.

### 2.4. Data Processing and Analysis

Socio-demographic and vaccination history data from each country were merged into cross-country datafiles and analyzed for frequency distributions and to establish RTS,S uptake status (software used for quantitative data: Excel version 2310, SPSS version 29.0.1.0, Tableau version 2021.2). Vaccination history data were treated as missing if the child health card was not seen at R3. In this paper, the following terms describe RTS,S uptake status:

Non-adoption: zero doses received.

Adoption: one [RTS,S-1] or two [RTS,S-2] doses (“one-” and “two-dose cases”) received.

Continuation: three [RTS,S-3] doses (“three-dose cases”) received.

Completion: four [RTS,S-4] doses (“four-dose cases”) received.

Audio recordings were translated into English, transcribed in full, and transcripts were merged into round-specific NVivo 12 Pro [17] projects. Transcripts were coded after each round by two cross-country coders (JP, NG). Initial coding schemes to identify uptake facilitators and barriers were iteratively refined to include inductively identified themes. Verification of coding consistency and ensuring consensus on definitions was performed weekly, or more frequently, throughout the analysis processes.

Within- and across-case analysis [18] was performed on individual caregiver data to reveal commonalities and variations within the cohort and changes over time. Across-case analysis involved creating case-by-theme matrices [19], with select themes summarized numerically to display the patterns [20] found at each round and to detect pattern changes across the three rounds. Within-case analysis involved developing individual case summaries to identify the main incidents and factors that appeared to influence RTS,S uptake. Case-by-summary matrices facilitated across-round comparison and longitudinal coding [14,21].

The HUS’s qualitative description complements inferential analyses on data from representative household surveys conducted as part of the pilot evaluations [5].

## 3. Findings

At R1, 188 caregivers were enrolled in the study; 36 were subsequently LTFU, with 10 individuals replaced in Kenya at R2, bringing overall enrollment to 198 and generating a final sample of 162 at R3 (Table 1).

All but six (*n* = 6/162) caregivers in the R3 sample were mothers of the RTS,S-eligible child. Most were between 19–34 years old, married or cohabitating, had completed primary school or more, and had more than one child (Table 2). Eighty-one RTS,S-eligible children were female, eighty were male, and there was one case with missing child gender data.

### 3.1. RTS,S Uptake

At R3, 152/162 (94%) child health cards were available to extract vaccination history from. Unless otherwise specified, the findings presented in this section focus on these 152 cases (Table 3).

As shown in Table 3, 64% (*n* = 98/152) of children received all four RTS,S doses and 93% received at least one dose. Table 4 lists the main facilitators and barriers to RTS,S uptake discovered through coding.

### 3.2. Uptake Facilitators

#### 3.2.1. A Trajectory of Trust

Corresponding to definitions of vaccine trust [22], caregivers at R1 expressed confidence in childhood vaccination overall (“measles and polio are no longer around because of vaccines”), government intentions (“the government can’t bring anything bad”), and health worker expertise (“health workers know best”). Nonetheless, many caregivers at this time had limited awareness of the new malaria vaccine, asking about its purpose and “why it has come out now”. Others had concerns about too many vaccines, more injections, or about RTS,S’s origins, safety, and rumored investigational status, and some participants were confused about why the vaccine was not given nationwide:

“Why [is] something that will help children being given in only three regions”? (G_C5_002_R1)

Despite such apprehensions, RTS,S adoption was high, with 77% (*n* = 140/183) of cohort children having received one or more doses at R1. The acceptance of the initial doses was tightly linked to caregivers’ trust in vaccines and the system:

“The concern I first had was because people were saying this new malaria vaccine is bad, that they were researching it… I got to a point when I let this concern go, because I know when hospitals come with an intervention, they’ve tried it out and start giving it after it’s certified to be good”. (M_C21_021_R1)

“They said there’s going to be a new malaria vaccine and we mothers shouldn’t panic… The nurses have gone to school and know what they’re talking about. So we became calm and accepted it”. (G_C6_002_R1)

By R2, the proportion of caregivers who adopted RTS,S increased to 141/165. Specific confidence in RTS,S also prominently emerged at this time, almost universally predicated on the perceived benefits (“she’s not getting sick as often”) and safety (“nothing bad happened”) of the vaccine. Confidence in RTS,S deepened by R3. Regardless of the number of doses the child had received, almost all caregivers at R3 believed that malaria occurred less frequently and, when it did occur, was less severe in vaccinated children:

“When I go to the hospital, I only see adults now, not children”. (M_C25_047_R3, child received 0 doses)

“If I sit down and analyze things, I know that malaria in my child is not severe like it used to be”. (G_C4_003_R3, child received 4 RTS,S doses)

Partly due to an increased focus on RTS,S in the R2 and R3 interviews, the trajectory of trust in RTS,S was striking in its consistency within and across countries. Text Box 1 exemplifies the trajectory in an individual caregiver and Figure 3 displays the pattern in the samples from each country. Each representing an individual, the blue dots in Figure 3 signify that the caregiver expressed general vaccine trust one or more times during the interview (e.g., R1 in Text Box 1); red dots signify expressions of specific trust in the new malaria vaccine (e.g., R2 and R3 in Text Box 1). Rather than a shift from general to specific trust, these findings reveal instead a dynamic linkage between the two. While foundational trust in vaccines/the system enabled initial acceptance, even with trepidation (R1), the observed safety and benefits of RTS,S over time (R2 and R3) lead to its inclusion within the broader family of trusted childhood immunizations. (Appendix A).

Box 1Typical RTS,S trust trajectory (K_C18_002).R1—“If the government approved something I’ll go for it. I don’t sit back and question it. The government has good reasons for bringing any vaccine”.R2—“I haven’t heard any problems due to the vaccine… [and]… my understanding [of RTS,S] is, okay, greater since I started taking my child to receive it. This vaccine has helped us… It has helped me a lot. I have not been using money visiting the hospital all the time”.R3—“My child has not been sick, and he is now two years old. If he had not been vaccinated, he would have been sick twice or thrice by now”.

#### 3.2.2. Four-Dose Cases: Reasons for Completing the Schedule

By R3, 64% (98/152) of children for whom we had complete data had received all four RTS,S doses (Table 3). Importantly, nearly a third of these children’s caregivers (*n* = 29) described initial hesitation (“my heart hasn’t welcomed it completely”) in accepting RTS,S-1. Foundational vaccine trust, sometimes coupled with additional facilitators, helped these 29 caregivers overcome their initial concerns:

“When they told us about the vaccine we didn’t understand. This made my mother-in-law forbid us from taking the child, fearing [RTS,S] might kill her. Then the child’s father told me to take her since it’s the nurses who brought the vaccine”. (G_C1_002_R1, child received all four doses)

Fourth-dose prompts were also common, including encouragement from family and peers, the use of the child health book to remember vaccination visit dates, reminders from health workers and, in a few instances, interaction with HUS staff. Overwhelmingly, however, the impetus to complete the schedule was driven by the perceived vaccine benefits, frequently expressed in comparison to unvaccinated children or previous health status in the vaccinated child:

“She used to get sick almost every month. After the first dose it started reducing slowly to the second then the third and fourth. From January she has not been sick”. (K_C13_006_R3)

Positive sentiments about RTS,S were also pronounced in caregivers whose children received 0–3 RTS,S doses. Other barriers prevented these caregivers from adopting RTS,S or completing the schedule.

### 3.3. Barriers to Adoption, Continuation, or Completion

Figure 4 displays the barriers contributing to 54 children receiving zero or fewer-than-four RTS,S doses by R3. Each column in the figure represents one individual. Colored cells going down the column indicate barriers (listed in column 1) coded to that individual’s data. Gray cells indicate that no barriers were coded.

The barriers across uptake categories and countries were generally similar, but a few differences are highlighted below.

#### 3.3.1. Three-Dose Cases: Reasons for Non-Completion

Caregivers of three-dose children (*n* = 34) cited fewer barriers on average (1.53/caregiver) compared to zero- and one-/two-dose cases (Figure 4). Without prompting, three of these individuals, represented in the last three columns of the figure, described specific and imminent plans to go for RTS,S-4. No barriers were discerned in these cases. Five other caregivers from Malawi were adamant that the child received RTS,S-4, supporting their claims with visit details:

“I told the doctor my child hadn’t received his last dose, so they jabbed him and forgot to write it down. There were a lot of people and they were working fast because of Corona. After the jab they said he finished the doses”. (M_C27_062_R3)

It is possible that these five children received RTS,S-4, but we retain them as three-dose cases and code the inconsistency as a health system barrier (Figure 4), indicating the weakness in record-keeping or communication. No other fourth-dose barriers were described by these five caregivers.

A mix of health system, personal, attitudinal, and information barriers to RTS,S-4 were coded to the remaining 26/34 three-dose cases. Table 5 summarizes barriers for each case.

Singularly prominent barriers: Twelve (n = 12/34) caregivers of three-dose children described a singular or clearly prominent reason for missing RTS,S-4. Five cases involved access/availability issues (rows 1, 7, and 18, Table 5), including two instances where RTS,S-4 was not given because the child was sick and under treatment at the time (rows 6 and 10). A personal constraint—pregnancy, new baby, sickness—was the dominant reason given by three individuals (rows 5, 9, and 17). Two instances involved specific RTS,S-4 information gaps (rows 2 and 16) and two others complacency, including self-acknowledged “negligence” (row 14) and passivity (row 20):

“I was expecting she’d be vaccinated at the under-five clinic, so I thought it was finished… I did not ask”. (M_C23_030_R3)

Multiple interacting barriers: Fourteen (*n* = 14/34) caregivers of three-dose children cited multiple, often interacting barriers:

Ghana—travel interacting with low perceived need for RTS,S-4

“I know she was supposed to take four doses, but I travelled. If not for that I would have taken her. However, I think 3 doses are protecting her because she’s never been sick with malaria since she started the vaccine”. (G_C6_005_R3)

Kenya—information gaps interacting with health system and personal barriers

“When I was still giving birth, I just knew children only get one dose… Now they tell me there are four, others say three. I’m not sure what’s true… I thought at three he had completed. When [child’s] mother was here I told her to take him because he’s heavy for me. She went but found the clinic closed [strike] and was told to go to a private facility. She went there and was told there was no malaria vaccine. Then [child’s mother] traveled… I got tired and don’t even know where that private hospital is”. (K_C15_003_R, child’s grandmother)

Malawi—COVID closure interacting with competing livelihood demands

“He always receives [vaccination] in good time. Only one [RTS,S dose] is remaining but when I went, I was told to come next month. I’ve been busy with planting season, so I left that on hold”. (M_C24_040_R3)

All three children in these examples were eligible to receive RTS,S-4 at R3, according to country guidance. Whether or not they eventually received the final dose, the examples underscore the importance of timely service access. Once the opportunity to receive RTS,S-4 is missed due to a lack of service it is uncertain whether the caregiver can or will make the effort again.

Country-specific barriers to completion: As summarized in Table 5, in Ghana, missing vaccination visits while traveling was a notable RTS,S-4 barrier. In Kenya, a health worker strike and stockouts were cited by 5 of 12 caregivers from Kenya. In Malawi, which opted for a “silent introduction” of RTS,S, persistent information gaps were observed, often overlapping with limited caregiver engagement. Eleven of the eighteen three-dose cases from Malawi showed limited awareness, limited engagement, or both:

“As parents we don’t count how many doses she received today, or anything. No, once we give birth, we leave the child in the hands of health providers. Anything to do with their health is to be done by them”. (M_C23_029_R3)

Across all the countries, by R3 all but a few three-dose children were under 36 months old and eligible to receive RTS,S-4, according to country guidance. Caregiver awareness of RTS,S-4 eligibility up to three years of age is unclear from our data.

#### 3.3.2. One-/Two-Dose Cases: Reasons for Non-Continuation

Eight children received two RTS,S doses by R3 and one child only one dose. The continuation barriers described by caregivers of these children were similar in nature and pattern to three-dose cases. Six caregivers had access barriers: COVID closures (two cases in Malawi), a health worker strike (two cases in Kenya), traveling (one case in Ghana), and distance from the service (one case in Kenya). Personal circumstances (“busy farming”) were cited in four instances and complacency was implied or explicit in five cases, often compounded by other issues (Text Box 2). Information gaps were also common (“What is it for?”), sometimes directly interfering with dose continuation.

Box 2Interacting barriers to RTS,S continuation (M_C27_058).R1—Caregiver expresses vaccine trust but is concerned about the number of vaccines and pain from injections. She has skipped vaccination visits in the past.R2—Caregiver indicates confidence in RTS,S but appears unmotivated and has farming demands. Additionally, under-five clinics were suspended due to COVID, causing her to miss the RTS,S-3 visit.R3—Repeating explanations from prior rounds, the caregiver adds: “I just chose to stop at the second dose. I was unmotivated for my child to get another jab, so I stopped there.” The child subsequently suffered from malaria and the caregiver wonders if this could have been avoided if she’d finished the doses. She further explains: “When I stopped last year, we weren’t allowed to go to under-5 [clinic] because of COVID. Then they said my child was too old, so I just stopped going.”.

Two differences between one-/two-dose and three-dose cases can be identified. First, while fears about AEFIs, more injections, the number of vaccines, or RTS,S rumors were absent or inconsequential in three-dose cases, five of the nine caregivers of one-/two-dose children had these concerns (Figure 4 and Text Box 2). Second, caregivers of one-/two-dose children described twice as many barriers on average (3.22/caregiver) compared to three-dose cases. Apart from one caregiver, who discontinued the doses due to church opposition, all other one-/two-dose cases described multiple issues likely interacting to obstruct continuation with the schedule.

Of note, eight out of nine one-/two-dose children were under-immunized for other childhood vaccines (Table 3). Additionally, the vaccination cards of two children showed the child had received the measles-1 vaccine, which would have been delivered at the same visit as RTS,S-3. We cannot tell from the data why these two children received measles-1 but not RTS,S-3.

#### 3.3.3. Zero-RTS,S-Dose Cases: Reasons for Non-Adoption

Eleven children at R3 had not received any RTS,S doses according to their child health cards; five were also under-immunized for other childhood vaccines (Table 3). Caregivers of these children describe multiple (2.73/caregiver) uptake barriers, but a core reason for non-adoption was apparent in the data from all but one case (Table 6).

Adverse events and fears about safety: Concerns about safety and side effects were the principal reasons for non-adoption among caregivers #1–4 (Table 6). Two individuals from Ghana (#1–2) had RTS,S-specific concerns; both expressed trust in “the old vaccines, but not this new one” and emphasized that their husbands were also RTS,S-hesitant. After initially refusing RTS,S, the caregivers’ perceptions changed by R2, #1 describing the rumors as “just hearsay”, noting that RTS,S-vaccinated children were “still walking around here”. Case #2, herself a health worker responsible for compiling malaria reports, offered this evidence-based explanation for her change of heart:

“I know how malaria cases used to be compared to now… Because of how cases are reducing, I thought I made a mistake. To be sincere, malaria has reduced”. (G_C2_007_R3)

Both these caregivers sought RTS,S-1 at the nine-month child health visit. Caregiver #2 was told it was too late while caregiver #1 believed the child received a dose:

“The nurse told me, and I saw him give two [injections] on one leg and one on the other”. (G_C2_006_R3).

Cases #3 and #4 (Table 6) involved hesitancy due to AEFIs from other vaccines. Despite her own acceptance of RTS,S and health worker attempts to persuade her husband to agree, #3 explains that “he wouldn’t allow me to take her”. In contrast, case #4 struggled with her own ambivalence:

R1—“They [twins] cry so you end up confused… it takes away some of the joy [getting the child vaccinated]”.

R2—”I haven’t taken them because I feared it would worsen [prior AEFIs]… I told myself I’ll just continue because it’s not too late”.

R3—“I decided against it… because of the abscess… I was afraid. Nothing else stopped me. Later insisting: I was committed somewhere else and not able to make it”. (K_C11_004)

Access barriers: Three caregivers from Kenya (#5–#7, Table 6) focused on access issues. Individual #7 lived far from the facility providing RTS,S and took her child to a close-by private facility for vaccinations. This caregiver remained under-informed about RTS,S, asking the interviewer at R2, “Where can I go to get it?” Caregiver #6 explained that she tried to get the child vaccinated with RTS,S at three different facilities but was told each time that it was out of stock. By the time the stockout was resolved it was too late. However, having heard “other mothers praise the vaccine”, she made sure her next-born child received RTS,S. Caregiver #5 had a more complex relationship with the health system, complaining about health workers “who don’t like people” and being “told [about] a door-to-door campaign” that never transpired: “Once I was told that, I just stayed put waiting for them, but we haven’t seen anyone yet”.

Information gaps: Four caregivers from Malawi (#8–#11) exhibited persistent RTS,S information gaps. Reasons for non-adoption could not be determined for caregiver #8; all three others (#9–#11) believed that their children had received one or more RTS,S doses and recalled details about the vaccination visit:

R1—The caregiver says that to reject vaccines is “dangerous” and is enthusiastic about RTS,S. She understands little about RTS,S and believes her child has already received three doses.

R2—The child was sick with malaria since R1, which she attributes to not using a bed net (“they were torn”). Still under-informed about RTS,S, the caregiver asks: “How many [doses] does he have left”?

R3—Caregiver remains under-informed and adds: He received the vaccine, but I don’t know where they wrote it… The health worker told me it was for malaria, but it doesn’t mean he will never suffer from malaria again. (M_025_047)

Country-specific barriers to adoption: Access/availability issues were notable adoption barriers in Kenya. In Malawi, information needs and possible record keeping errors were prominent. The situation in Ghana was different. Wide exposure to early rumors (affecting more than half of Ghanian participants at R1) stands out in zero-dose cases. Despite rumors easing over time, rumor-caused acceptance delays, coupled with apparent health worker confusion about RTS,S-1 eligibility, resulted in children missing all their doses.

## 4. Discussion

Qualitative findings from the HUS’s case-intensive examination of issues affecting RTS,S adoption, continuation, and completion, complement the results from quantitative studies [23] and add detail to prior formative research on the acceptability of a malaria vaccine. While formative studies were conducted in anticipation of a malaria vaccine [24,25,26,27] or in conjunction with a clinical trial [28], HUS data were collected contemporaneously with the provision of RTS,S through routine services. The study’s longitudinal approach permitted us to observe RTS,S acceptance as a dynamic process while its delivery through routine services provided real-world insights across diverse social and service settings.

Consistent with formative research [24,25,27,28], the child caregivers we interviewed emphasized the impact of malaria on their communities and the value of a malaria vaccine. They understood partial protection and appreciated the reduced disease severity as an important vaccine benefit. The caregivers’ own observations of reduced malaria frequency and severity reinforced their enthusiasm for RTS,S, possibly also strengthening their trust in child health services and vaccines more generally.

Importantly, our findings on the wide acceptance of RTS,S bring child caregiver voices to global discussions on the programmatic viability of a malaria vaccine with moderate efficacy [29]. They also inform demand creation and communications strategies. Highlighting the vaccine’s demonstrable benefits (e.g., through testimonials) while monitoring for, preempting, and addressing mis/disinformation, will help to achieve wide coverage with initial doses. Messages focused on completing the schedule, stressing that four doses provide the best protection for children under five, who are at the highest risk of dying from malaria, will address complacency as parents potentially become prematurely satisfied with the vaccine’s benefits.

Our case-intensive analysis of the reasons for non-adoption and missed doses point to specific opportunities to enhance adherence to a four-dose schedule.

Situation-specific communications: Our findings show that many missed doses were linked to service gaps. Whether due to temporary closures, stockouts, or schedule changes, caregivers who missed doses for these reasons were under-informed about what to do to keep their children up to date for RTS,S. Communicating when and where caregivers can bring their children when the service is unavailable will minimize such missed opportunities. Ideally done in advance of the interruption, communications about the service’s resumption, location, and dates would spare caregivers from making unnecessary trips and becoming frustrated with the performance of the health system. Where a short message service (SMS) is being used to promote vaccination [30], expanding SMSs to include these announcements could be effective.

Similarly, when a dose is not given to a sick child, caregivers need to know what to do once the illness resolves. However infrequent, such instances observed in our sample indicate an opportunity to strengthen provider training and service delivery protocols to prompt providers to provide clear and personalized instructions about bringing the child back to stay current with the schedule.

First dose interventions: Our findings suggest that greater clarity around RTS,S-1 eligibility is needed. Service interruptions, caregiver delays in accepting RTS,S, and travel away from home when the child first became eligible, were reasons for non-adoption. Improved awareness of the country’s RTS,S-1 cut-off age, among providers and caregivers alike, could avert these situations. RTS,S-1 eligibility should be emphasized in provider training and reinforced in reminders, updates/refreshers, and supportive supervision.

Fourth dose interventions: While uptake of the RTS,S in our sample was strong overall, a notable drop in adherence was observed after RTS,S-3, including among caregivers motivated to bring their children for vaccination. Alone or in combination, service barriers, competing life demands, imprecise understanding of RTS,S-4 eligibility, and satisfaction with the perceived protection from three RTS,S doses often outweighed or precluded the acceptance of RTS,S-4. Aligning the provision of RTS,S-4 with measles-2 could reduce missing the final doses for both series, especially if coupled with intensified messaging around the specific value of RTS,S-4. Caregivers’ understanding of the fact that RTS,S-4 could be given after two years of age was unclear in our data. Emphasizing the extended eligibility timeframe could result in more individuals finding time and motivation to bring their child for the final dose.

Preemptive interventions addressing AEFI fears related to vaccines: Fears of AEFIs, typically linked to prior experiences with other vaccines, led caregivers to refuse or default on RTS,S doses. Where feasible, flagging these caregivers as “AEFI concerned” and providing them with personalized education could reduce RTS,S hesitancy and refusals.

Child-health-book-centered interventions: Neglecting to use the child health books creates confusion and may lead to missed visits. Problems occur when caregivers do not understand or consult the books or when providers fail to update it. Interventions to integrate use of the child health book in clinical encounters could enhance accountability and adherence to the schedule. For example, upon administering a dose the provider could briefly educate/remind the caregiver about the book’s contents and how to use it to plan future visits. The caregiver would see that the book was updated and where the next appointment date was recorded. In establishing a kind of task-sharing to track doses, such interventions could result in improved data accuracy, caregiver engagement, and vaccination coverage.

Finally, our study revealed challenges for the phased subnational introduction of RTS,S. Subnational delivery and evaluations through the pilot studies fed into early rumors about the vaccine “being researched”. Although such suspicions did not endure, they contributed to RTS,S hesitation early on. Lack of access to the vaccine when caregivers traveled to locations outside of the pilot regions was another challenge, sometimes clearly linked to children missing doses. Given a dynamic RTS,S supply situation globally, and WHO guidance for phased introductions [31], it will be critical for newly introducing countries to anticipate these challenges in their vaccine introduction plans.

A few limitations of this study need mentioning. First, COVID interruptions changed the study context mid-course. The interruptions not only posed a barrier to RTS,S uptake/continuation, they also delayed R2 interviews by several months, potentially affecting caregiver recall or affecting the results in other indiscernible ways. Methodologically, while the longitudinal design provided useful insight into vaccine acceptance as a process, caregivers’ recurrent contacts with research staff heightened their awareness of the vaccine. The higher uptake of RTS,S in our cohort compared to coverage from routine immunization services for the same period [32] likely reflects these recurrent contacts. Finally, while findings from our qualitative sample have relevance for countries newly introducing the vaccine, we emphasize that they are not statistically representative and that generalization to broader populations should be avoided.

## Figures and Tables

**Figure 1 vaccines-11-01801-f001:**
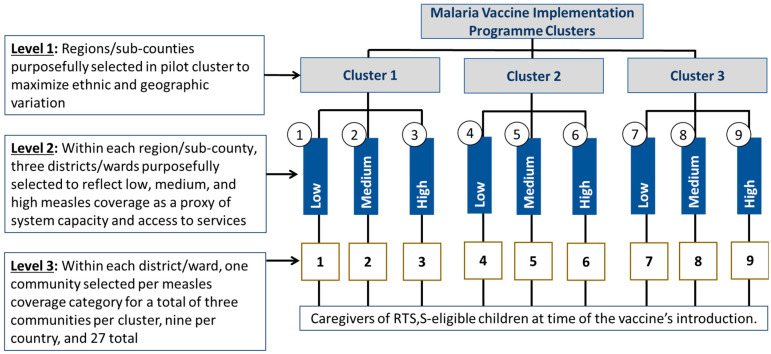
Study site sampling strategy used in each country.

**Figure 2 vaccines-11-01801-f002:**
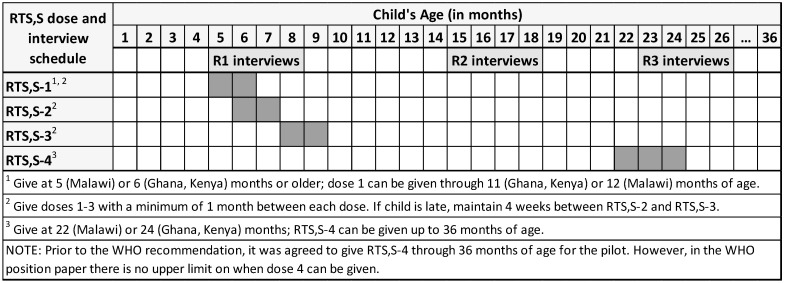
Country RTS,S schedules for the pilot and timing of interviews (conducted from around late 2019 to mid-2021).

**Figure 3 vaccines-11-01801-f003:**
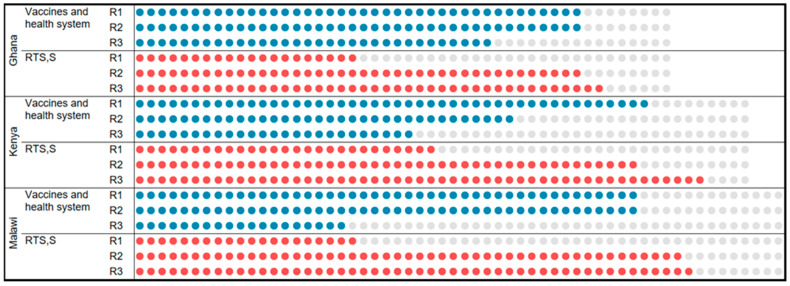
Caregiver expressions of general vaccine and RTS,S-specific trust by country and interview round. Blue dot: caregiver expressed general trust in vaccines or health systems one or more times during the interview. Red dot: caregiver expressed specific trust in RTS,S one or more times during the interview. Grey dot: no general or RTS,S-specific trust codes assigned.

**Figure 4 vaccines-11-01801-f004:**
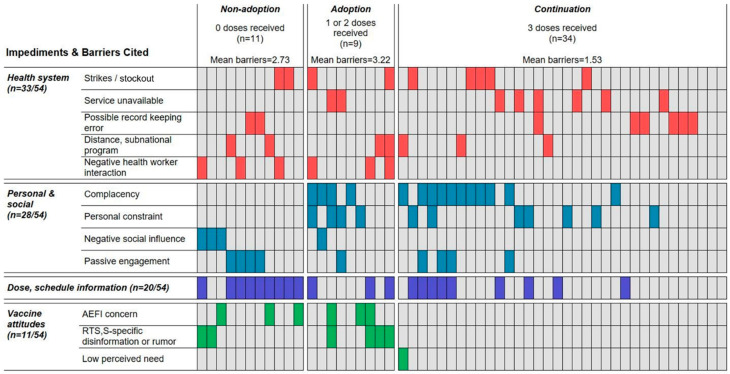
Barriers cited by caregivers (*n* = 54) whose children received zero, one or two, or three RTS,S doses. Columns represent individual caregivers. Colored cells going down the column indicate barriers coded to the individual’s data. Gray cells indicate no barriers were coded.

**Table 1 vaccines-11-01801-t001:** Enrollment, LTFU, and missing vaccination history by round.

	R1	R2	R3
Enrolled	188	188	198
Replaced	0	10	0
Total number enrolled	188	198	198
LTFU	0	−25	−36
Total number interviewed	188	173	162
Vaccination card not seen	−5	−8	−10
Cases with complete data for uptake analysis	183	165	152

**Table 2 vaccines-11-01801-t002:** Caregiver characteristics at R 3 by country.

Characteristic		Number (%)		
		Ghana (*n* = 49, 13 LTFU)	Kenya (*n* = 55, 18 LTFU and 10 Replaced)	Malawi (*n* = 58, 5 LTFU)
Sex	Female	47 (95.9)	51 (92.7)	58 (100.0)
	Missing	--	1 (1.8)	--
Age (years)	15–18	1 (2.0)	1 (1.8)	--
	19–24	10 (20.4)	10 (18.1)	21 (36.2)
	25–29	13 (26.5)	15 (27.2)	14 (24.1)
	30–34	14 (28.6)	12 (21.8)	10 (17.2)
	35–40	8 (16.3)	10 (18.1)	10 (17.2)
	40+	3 (6.1)	6 (10.9)	3 (5.1)
	Missing	--	1 (1.8)	--
Marital status	Married or cohabiting	39 (79.6)	52 (94.5)	45 (77.5)
	Divorced, widowed, unmarried	10 (20.4)	2 (3.6)	13 (22.4)
	Missing	--	1 (1.8)	--
Education (highest completed)	None	4 (8.2)	--	4 (6.8)
	Primary	9 (18.4)	37 (67.2)	45 (77.5)
	Secondary	31 (63.3)	13 (236)	9 (15.5)
	Post-secondary	5 (10.2)	4 (7.2)	--
Number of children	1	6 (12.2)	4 (7.2)	18 (31.0)
	2	12 (24.5)	8 (14.5)	12 (20.6)
	3+	31 (63.3)	42 (76.3)	28 (48.2)
	Missing	--	1 (1.8)	--
Relation to child	Mother	48 (98.0)	50 (90.9)	58 (100)
	Grandparent or other	1 (2.0)	4 (7.2)	--
	Missing	--	1 (1.8)	--

**Table 3 vaccines-11-01801-t003:** RTS,S doses and other childhood immunizations received by RTS,S-eligible children at R3.

Data Completeness	RTS,S Doses Received				BCG, Penta, and Measles Immunization Status	
	Doses Received	N	% of Total Enrolled (*n* = 198)	% of Cases w/Complete Data (*n* = 152)	Fully Immunized (*n* = 115/152)	Partially Immunized (*n* = 37/152)
Complete (N = 152)	4	98	49.5%	64.5	87	11
	3	34	17.2%	22.4	20	14
	2	8	4.0%	5.3	1	7
	1	1	<1%	<1	0	1
	0	11	5.6%	7.2	7	4
Incomplete (N = 46)	Card not seen	10	5.1%			
	LTFU	36	18.2%			
Total		198	100%			

Fully immunized refers to children who received BCG, Penta-1, Penta-2, Penta-3, MR-1, and MR-2 as of the R3 interview. Partially immunized indicates one or more of these vaccinations were missing from the child health card.

**Table 4 vaccines-11-01801-t004:** Main facilitators and barriers to RTS,S uptake.

Category	Main Themes Discovered	
	Facilitators	Impediments and Barriers
Health System	Positive interaction with health worker, positive experience during child health visit. Reminders from health workers, use of/reliance on child health book. Encouragement and support from (community) health worker.	Health worker strikes. Vaccine stockouts. Service unavailable (turned away, rescheduled, COVID interruption). Distance from the service site (including subnational program challenges). Record-keeping issue.
Personal and Social	Encouragement and support from personal network member. Supportive social norms (e.g., “We just take our children when we’re told”).	Complacency, dose/vaccination not prioritized, lack of caregiver motivation or engagement to make the vaccination visit. Personal constraint (livelihood demands, social obligations). Negative social influence (partner refusal/discouragement, religious restrictions).
Information and Knowledge	Awareness of a new malaria vaccine and what it is for. Informed, knowledgeable about RTS,S eligibility and dose schedule.	Limited information about or awareness of a malaria vaccine generally. Lack of information or confusion about RTS,S schedule, timing of doses, and child eligibility.
Perceptions and Attitudes	Confidence in vaccine safety and efficacy, the health system, and health workers’ skills and knowledge. Trust in the intentions of the government, industry, and others (e.g., researchers) involved in vaccine development and promotion. Perceived need for a malaria vaccine and/or specific benefits perceived from the uptake of RTS,S.	RTS,S-specific disinformation and rumors. Negative clinical encounter or interactions with health workers. AEFI concerns and/or past AEFI experience (perceived or real). Low perceived need for a malaria vaccine.

**Table 5 vaccines-11-01801-t005:** Specific barriers to RTS,S-4 uptake *.

Country and ID		Row	Service (Access, Availability)	Personal (Constraint, Attitude)	Information (Awareness, Knowledge)
Ghana	C4_004	1	Traveling; away from home		
	C5_004	2			Unaware of 4th dose
	C6_004	3	Clinic too far away	“I don’t plan on taking her” (No time)	
	C6_005	4	Traveling during RTS,S-4 visit	“I think the 3 doses are protecting her”	
Kenya	C13_002	5		“I was pregnant and very tired”	
	C13_004	6	RTS,S-4 withheld; child being treated		
	C14_007	7	Health worker strike, stockout		
	C15_003R	8	Health worker strike, stockout	Limited mobility (grandmother)	Limited awareness of doses and schedule
	C16_003	9		With new baby: “I was kind of tired”	
	C16_004	10	RTS,S-4 withheld; child being treated		
	C17_007	11	Health worker strike	Has not returned since strike ended	
	C18_002	12	Health worker strike	With the strike, “I developed some negligence”	
	C18_005	13	Health worker strike	Has not returned since strike ended	
	C18_007	14		“I’m remaining with one; have been negligent”	
Malawi	C20_009	15	COVID closure, then discouraged		Confused about number of doses
	C20_010	16			Confused about doses and the number child has received
	C21_015	17		Has been sick and has competing demands	
	C22_028	18	Turned away; schedule change		
	C23_029	19		“As parents, we don’t count the doses”	Limited awareness of doses and schedule
	C23_030	20		Attends U5 clinic, assumes fully vaccinated	
	C23_031	21		“I fell off” in those months.	Limited awareness of doses and schedule
	C23_033	22		“We don’t ask [or] count”	Surprised to learn there are four doses
	C24_038	23		Attends U5 clinic, assumes fully vaccinated	Limited awareness of doses and schedule
	C24_040	24	COVID closure	“I’ve been busy with planting season”	
	C25_049	25		Too busy to attend last U5 clinic	Unaware that child is missing RTS,S-4
	C27_061	26	COVID closure and record error		

* Table excludes three cases where caregivers described clear plans to go for RTS,S-4 and five cases where the caregivers believed the child received RTS,S-4 and offered details of the fourth-dose visit.

**Table 6 vaccines-11-01801-t006:** Main barriers to adopting RTS,S-1 and immunization status of the zero-RTS,S-dose children for other vaccines.

Case		Country	Key Adoption Barrier		Main Reason for Non-Adoption of RTS,S-1	Immunization Status for Other Vaccines
			Main Barrier	Reinforcing Barrier(s)		
Hesitancy	1	Ghana	Hesitant (RTS,S rumors)	Partner hesitant (RTS,S rumors) Info needs	Delayed acceptance	Missing BCG
	2	Ghana	Hesitant (RTS,S rumors)	Partner hesitant (RTS,S rumors)	Delayed acceptance	Complete
	3	Ghana	Partner refusal (past AEFI)	Partner’s religious beliefs	Partner refusal	Complete
	4	Kenya	Hesitant (past AEFI)	RTS,S info needs Distance from facility Personal constraints	AEFI fears	Missing Penta3 and measles series
Health System	5	Kenya	HW strike, stockouts	RTS,S info needs Poor HW attitudes	Service unavailable	Missing Penta3 and MR2
	6	Kenya	Stockouts	RTS,S info needs	Service unavailable	Complete
	7	Kenya	Distance from facility; uses private clinic closer to home	RTS,S info needs	Service inaccessible	Complete
Info Needs	8	Malawi	RTS,S info needs	Fears (too many injections)	Unclear	Missing BCG
	9	Malawi	RTS,S info needs	Poor HW attitudes	Believed child rcv’d some RTS,S doses	Missing Penta3 and measles series
	10	Malawi	RTS,S info needs	RTS,S info needs	Believed child rcv’d RTS,S-1	Complete
	11	Malawi	RTS,S info needs	RTS,S info needs	Believed child rcv’d some RTS,S doses	Complete

1. G_C2_006; 2. G_C2_007; 3. G_C5_006; 4. K_C11_004; 5. K_C11_005; 6. K_C14_004; 7. K_C18_004; 8. M_C20_013; 9. M_C23_032; 10. M_C25_046; 11. M_C25_047.

## Data Availability

In addition to data contained within the article, additional data are available in the Appendix A including the quantitative dataset on socio-demographics and child vaccination history and qualitative summaries (from R1 to R3) on each under-dose case and qualitative replies from all cases at R3 to malaria frequency and severity statements.

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
