# Peer review of "Acceptance of and Adherence to a Four-Dose RTS,S/AS01 Schedule: Findings from a Longitudinal Qualitative Evaluation Study for the Malaria Vaccine Implementation Programme"

_vaccines, 2023, doi:10.3390/vaccines11121801_

Round 1

Reviewer 1 Report

Comments and Suggestions for Authors

High quality manuscript describing the outcomes of a healthcare utilization study for the malaria vaccine. Mo major concerns or significant feedback.

If possible, word choice in the abstract could be improved. Numbers refer to interview rounds, vaccine doses, and children. Close reading is needed to keep them straight.

The connection between RTS,S uptake and general vaccination uptake (table 3) is striking. I find the responses in table 5 (specific barriers to RTS,S-4) tremendously insightful. Thank you for this work.

Author Response

Thank you for your feedback and comments. We have revised the abstract to make references to interview rounds and number of doses more distinct and clearer.

Reviewer 2 Report

Comments and Suggestions for Authors

This is a very extensive and qualitative description of caregivers impressions on a four dose vaccine schedule for malaria prevention in three subsaharian regions. The authors opts for a extensive description of the impressions of caregivers, without any statistical approach. This is also an effect of the fact that the caregivers impressions affects very little the completion of vaccine schedule. This fact was causes by the high acceptance of the vaccine by caregivers.

This is the main problem of the article as they had few comprehensive conclusions on their data, being merely descriptive or anedoctal in most cases. 

Despite those facts, if the editor had no space problems for publication, the manuscript is adequately written and also presented all of their data on a complex four dose vaccine acceptance in a subsaharian area.

If they perform the adequate statistical comparisons of some data, as the three regions demography and motherhood data, showing if there are differences in education level, age group and childrens in families; or the % of acceptance after each dose, as shown in the colored dot scheme, the manuscript would be acceptable. 

Author Response

Thank you for your feedback and comments. We appreciate your observation about the absence of statistical analyses on our study’s data. The essential purpose of the study, however, was descriptive in nature, to understand and describe factors that shape the vaccine’s uptake (lines 68-71) in diverse community settings where RTS,S was being provided as part of the pilots. While our purposive sampling criteria and approach was systematically used across the three countries to select 27 study sites and child caregivers in the cohort, the sample is not representative and therefore not appropriate for inferential statistics. We have, however, added short text to mention our qualitative study as part of the RTS,S pilot evaluations was intended to complement behavioral study results from representative survey samples (lines 47-48)

Reviewer 3 Report

Comments and Suggestions for Authors

The authors have conducted a large-scale qualitatitve investigation into the perception and uptake of the RTS,S vaccine. The article is well written and well organized. The findings are extremely important to the malaria and vaccine communities. I only have some minor suggestions.

1)    In the abstract the second sentence (lines 9-10) is incomplete. 

2)    In the abstract the authors should state the size of the cohort at the beginning of the study in the third sentence (line 10) 

3)    It’s not needed for publication in my view, but the article would be strengthened with a visual conceptual framework of the findings. Would the authors be willing to pull together what they learned into a visualization? Perhaps they have already done this for communication with partners.

Author Response

  • In the abstract the second sentence (lines 9-10) is incomplete.

Thank you for this observation. We’ve completed the sentence in the abstract accordingly.

  • In the abstract the authors should state the size of the cohort at the beginning of the study in the third sentence (line 10).

We’ve added the total number of caregivers enrolled in cohort in this sentence.

  • It’s not needed for publication in my view, but the article would be strengthened with a visual conceptual framework of the findings. Would the authors be willing to pull together what they learned into a visualization? Perhaps they have already done this for communication with partners.

Thank you for this suggestion. We have been working with communications experts to integrate some of this study’s findings into various guidance and materials to support newly-introducing countries. However, these do not fully reflect the findings from the HUS. While we don’t have time to develop a conceptual framework of the findings in time to submit this revise-and-resubmit, we appreciate the idea and, if the paper is accepted and the journal permits it, we will develop a summary visualization of findings that we could include as a supplement. Again, thank you for the suggestion.

Reviewer 4 Report

Comments and Suggestions for Authors

Thank you for this very interesting study especially given considerable hesitancy towards COVID-19 vaccines in children across Africa and wider, as well as vaccine hesitancy generally fueled by misinformation surrounding the COVID-19 vaccine and others. My comments are only minor as I do believe your approach and findings were very comprehensive. These include:

A) Introduction - May be worth saying that in Africa routine immunisation of children was appreciably affected by lockdown and other measures associated with the COVID-19 pandemic (Abbas K et al. Routine childhood immunisation during the COVID-19 pandemic in Africa: a benefit-risk analysis of health benefits versus excess risk of SARS-CoV-2 infection. Lancet Glob Health. 2020;8:e1264-e72; Causey K et al. Estimating global and regional disruptions to routine childhood vaccine coverage during the COVID-19 pandemic in 2020: a modelling study. Lancet. 2021;398:522-34; Gaythorpe KA et al. Impact of COVID-19-related disruptions to measles, meningococcal A, and yellow fever vaccination in 10 countries. Elife. 2021;10) - so good to get back on track with an effective malaria vaccine that appears reasonably well accepted 

B) Methodology - I have no disagreements with the methodology especially the longitudinal approach, etc., over a long period of time. However good to know:

a) The time period when the study was conducted - and was any part affected by lockdown measures for COVID-19?

b) How was the questionnaire developed? Was this based on any publications/ considerable experience of the co-authors (sorry if I missed this information) 

C) Discussion - I would like to know more about potential advice to Governments, etc., as they role out the vaccine to further enhance acceptance rates. I know the caveats - but you must have some ideas about potential activities given your findings.  

Author Response

  1. Introduction - May be worth saying that in Africa routine immunisation of children was appreciably affected by lockdown and other measures associated with the COVID-19 pandemic.

Thank you for this suggestion and for the citations. We’ve had time to read two of them and have referenced them in to updated text (lines 43-45) to reflect your suggestion. And we agree! “so good to get back on track with an effective malaria vaccine that appears reasonably well accepted.”

  1. Methodology - I have no disagreements with the methodology especially the longitudinal approach, etc., over a long period of time. However good to know:
  2. a) The time period when the study was conducted - and was any part affected by lockdown measures for COVID-19?

The studies in each country were slightly different depending on the RTS,S launch dates, but overall, we began some data collection in late 2019 through early- to mid-2021. Round 2 interviews were a bit delayed due to COVID closures and a suspension of all research activities around April 2020. However, with approved mitigation plans, we were permitted to resume the interviews around July-August 2020 enabling us to maintain the interview timeline of collecting data between when the child should have received doses 3 and 4.

We’ve added short text to the Figure 2 title indicating the time range of when interviews were conducted (line 105) and added short text to lines 116-117 indicating that R2 interviews were slightly delayed to COVID closures but occurred nevertheless between doses 3 and 4 as planned.

  1. b) How was the questionnaire developed? Was this based on any publications/ considerable experience of the co-authors (sorry if I missed this information) 

Thank you for this comment. I’ve added the sentence (lines 121-122): Interview guides were developed to reflect key knowledge needs for the pilot evaluations and based on collective experience and expertise of study investigators.

  1. Discussion - I would like to know more about potential advice to Governments, etc., as they role out the vaccine to further enhance acceptance rates. I know the caveats - but you must have some ideas about potential activities given your findings.  

Thank you for this suggestion. We believe our discussion offers a fairly wide range of potential advice to governments – albeit succinctly given space constraints.

  • Lines 491-497 stress key demand generation messages.
  • Lines 502-511 provides more specific advice on how to avoid missed doses due to various reasons for service gaps (closures, cancellations, etc.), which can dampen demand.
  • Similarly Lines 513-517 provide specific advice on avoiding missed doses when a child is turned away/not given a vaccine (e.g., when sick), which is more of an information need, but can also dampen demand.
  • Lines 519-524 focus on key messages and ways (targeting health providers and caregivers) to clarify the time period when the child can receive the first dose, which will substantially increase the period that caregivers can accept the vaccine and that providers will initiate.
  • Lines 526-536 focus on interventions and messages that would increase acceptance and uptake of dose 4.
  • Lines 538-542 provides advice to intervene in especially vulnerable cases when a caregiver hesitates or refuses a dose due to past experiences with an AEFI.
  • Lines 544-553, though general in nature, advocates for promoting dose uptake by caregivers’ better understanding and utilization of the child health book to track the number of doses and remain aware of upcoming visits.

In sum, given the space constraints of the paper, we feel we have touched upon a fairly wide range of possible interventions, all of which are derived from what we observed in the data. If this is not satisfactory and the journal permits, we would be happy to consider specific changes that you recommend if the data support them.

Again, thank you for reviewing the manuscript and for your feedback.